# Genome-Wide Identification of the NF-Y Gene Family and Their Involvement in Bolting and Flowering in Flowering Chinese Cabbage

**DOI:** 10.3390/ijms241511898

**Published:** 2023-07-25

**Authors:** Zhehao Jiang, Yuting Wang, Wenxiang Li, Yudan Wang, Xiaojuan Liu, Xi Ou, Wei Su, Shiwei Song, Riyuan Chen

**Affiliations:** College of Horticulture, South China Agricultural University, Guangzhou 510642, China; zhjiang@stu.scau.edu.cn (Z.J.); wangyt@stu.scau.edu.cn (Y.W.); wxli@stu.scau.edu.cn (W.L.); ydwang@stu.scau.edu.cn (Y.W.); liuxjjy628@stu.scau.edu.cn (X.L.); ouxi@stu.scau.edu.cn (X.O.); susan_l@scau.edu.cn (W.S.); swsong@scau.edu.cn (S.S.)

**Keywords:** flowering Chinese cabbage, NF-Y gene family, gibberellin, bolting

## Abstract

Flowering Chinese cabbage (*Brassica campestris* L. ssp. *Chinensis* var. *utilis* Tsen et Lee) is a widely consumed vegetable in southern China with significant economic value. Developing product organs in the flowering Chinese cabbage involves two key processes: bolting and flowering. Nuclear factor Y (NF-Y) is a heterotrimeric transcription factor known for its crucial role in various plant developmental processes. However, there is limited information available on the involvement of this gene family during flowering during Chinese cabbage development. In this study, 49 *BcNF-Y* genes were identified and characterized along with their physicochemical properties, gene structure, chromosomal location, collinearity, and expression patterns. We also conducted subcellular localization, yeast two-hybrid, and transcriptional activity assays on selected *BcNF-Y* genes. The findings of this study revealed enhanced expression levels of specific *BcNF-Y* genes during the stalk development and flowering stages in flowering Chinese cabbage. Notably, BcNF-YA8, BcNF-YB14, BcNF-YB20, and BcNF-YC5 interacted with BcRGA1, a negative regulator of GA signaling, indicating their potential involvement in GA-mediated stalk development. This study provides valuable insights into the role of *BcNF-Y* genes in flowering Chinese cabbage development and suggests that they are potential candidates for further investigating the key regulators of cabbage bolting and flowering.

## 1. Introduction

Flowering Chinese cabbage (*Brassica campestris* L. ssp. *Chinensis* var. *utilis* Tsen et Lee) is a highly cultivated and productive vegetable in southern China. It belongs to the Chinese cabbage subspecies of *Brassica* species in the *Brassicaceae* family. The main product organ is the flowering stalk. The formation of the flowering stalk involves two simultaneous processes: bolting, which is characterized by stem elongation and thickening, and flowering. Gibberellin (GA) is a crucial factor influencing bolting in flowering Chinese cabbage. Treatment with exogenous gibberellin A3 (GA3) has been shown to induce the up-regulation of BcSOC1 and genes encoding cell wall structural proteins (*BcEXPA11*, *BcXTH3*) in flowering Chinese cabbage, thereby facilitating early bolting and flowering processes. Moreover, low temperatures trigger GA accumulation at the stem tip, accelerating growth and flowering. Conversely, using a GA synthesis inhibitor (PAC) suppresses the initiation of bolting and flowering [1,2,3,4]. *BcSOC1*, a key regulator in developing the flowering stalk, promotes early flowering and bolting when overexpressed. This is probably due to the upregulation of genes encoding cell wall structural proteins. In contrast, the knockdown of *BcSOC1* leads to significant phenotypic differences in plants, underscoring its importance in flowering Chinese cabbage [5]. Loss-of-function mutations in BcRGL1, a DELLA protein, affect the expression of various genes in flowering Chinese cabbage. These genes include the GA regulatory gene *BcGASA6*, flower-related genes *BcSOC1* and *BcLFY*, and genes encoding cell wall structural proteins *BcEXPA11* and *BraXTH3*. These mutations promote flowering and bolting. Conversely, overexpression of *BcRGL1* suppressed both bolting and flowering in flowering Chinese cabbage [6].

The nuclear factor Y (NF-Y) family, known as the CCAAT-binding factor (CBF) or heme-activating protein, is a conserved heterotrimeric transcription factor complex in eukaryotes. It consists of three subfamilies: NF-YA, NF-YB, and NF-YC. The NF-Y complex is formed by associating three members from different subfamilies [7,8,9]. NF-Y has been extensively studied and implicated in various biological processes, including plant stress responses, hormone signaling pathways, and seed development, among others [10,11,12,13,14,15,16,17,18]. For example, in rice, the complex formed by NF-YB1, NF-YC12, and bHLH144 binds to the *Wx* promoter, activating *Wx* transcription and regulating seed quality [19]. In poplar, PdNF-YB21 promotes root development by enhancing the expression of *PdNCED3*, a key gene involved in ABA synthesis, increasing ABA levels in the roots and promoting growth under drought conditions [12]. NF-YCs bind to the promoter of *BR6ox2* (a gene involved in BR biosynthesis) to inhibit BR biosynthesis. Furthermore, they maintain the stability of BIN2, a key inhibitor of the BR signaling pathway, thus influencing the photoregulation of hypocotyl elongation in *Arabidopsis* [20].

In recent years, the involvement of NF-Y family members in flowering and gibberellin signaling has gained significant attention. Studies have revealed that NF-Ys form a complex with CONSTANS to recognize key regulatory motifs in the promoter region of FT genes, thereby affecting flowering time [21,22,23,24,25,26,27]. In the photoperiodic and GA pathways, GA-mediated degradation of DELLA proteins releases NF-Y from the DELLA-NF-Y complex. This enables NF-Y to bind to NFYBE, a critical motif in the *SOC1* promoter, thereby promoting *SOC1* expression and accelerating flowering in *Arabidopsis* [28]. In *Arabidopsis*, the NF-YC-RGL2 module regulates seed germination by binding to a distinct CCAAT motif within the *ABI5* promoter, a key gene in the ABA signaling pathway. It also co-regulates a group of genes responsive to GA and ABA [29]. In roses, RhNF-YC9 regulates the petal expansion rate by influencing GA accumulation and expression of cell proliferation-related genes [30]. Moreover, DELLA proteins in *Arabidopsis* suppress *SOC1* expression by weakening the binding of NF-YCs to SOC1 through BRM. This promotes the binding of BRM to SOC1, collectively inhibiting flowering [31]. These findings highlight the crucial role of the NF-Y gene family in plant flowering and the GA pathway. Nevertheless, our understanding of the essential characteristics and functional significance of NF-Y transcription factors in the flowering Chinese cabbage is still limited. Therefore, this study aimed to investigate the role of the *NF-Y* gene family in the product organ development process of flowering Chinese cabbage.

Using bioinformatics methods, 49 *BcNF-Y* genes were identified in flowering Chinese cabbage, with 17 *BcNF-A*, 20 *BcNF-YB*, and 12 *BcNF-YC*. Comprehensive analyses were conducted to investigate various aspects of these genes, including their physicochemical properties, gene structure, motif composition, and conserved structural domains. Phylogenetic tree analyses were conducted using data from flowering Chinese cabbage, rice, and *Arabidopsis* to examine their evolutionary relationship. Additionally, collinearity analysis between flowering Chinese cabbage and *Arabidopsis* was performed to determine the potential roles of these genes. The expression patterns of *NF-Y* gene family members were examined in different developmental stages of flowering Chinese cabbage tissues and under GA treatment. Notably, several *NF-Y* family members that interact with DELLAs have been identified and further characterized for their subcellular localization and transcriptional activity. This study offers valuable insights into the involvement of the *NF-Y* gene family in the development of flowering Chinese cabbage. Further, it provides a good set of candidate genes for future investigations that can be explored in the future to identify key regulators of flowering Chinese cabbage development.

## 2. Results

### 2.1. Identification of BcNF-Y Family Members

A comprehensive analysis of *BcNF-Y* genes was performed in the flowering Chinese cabbage genome. This resulted in identifying 49 non-redundant *BcNF-Y* genes (17 *BcNF-YAs*, 20 *BcNF-YBs*, and 12 *BcNF-YCs*). These genes were named *BcNF-YA1* to *BcNF-YA17* (NF-YA subfamily), *BcNF-YB1* to *BcNF-YB20* (NF-YB subfamily), and *BcNF-YC1* to *BcNF-YC12* (NF-YC subfamily). Physicochemical analysis of the encoded proteins revealed significant variations in protein length. The length ranges from 102 (BcNF-YB2) to 1374 amino acids (BcNF-YA12). Additionally, the predicted molecular weight of the proteins varied, the lowest being 11670.12 Da (BcNF-YB11) and the highest being 155711.3 Da (BcNF-YA5). The theoretical isoelectric point (PI) of the proteins ranged from 4.27 (BcNF-YB8) to 10.11 (BcNF-YB19) (Table 1).

### 2.2. Analysis of Motifs and Conserved Domains in BcNF-Y Family Members

The MEME v5.4.1 software was used to identify motifs and analyze the motif composition of BcNF-Y proteins. The analysis revealed that members within the same subfamily exhibited similar conserved motifs (Figure 1a). In the BcNF-Y family, the BcNF-YA subfamily possessed a unique CBF domain, while the BcNF-YB and BcNF-YC subfamilies shared the CBFD_NFYB_HMF domain (Figure 1b). The presence of the same domain in BcNF-YB and BcNF-YC suggests functional similarities between these subfamilies. All members of the BcNF-YA subfamily contained motifs 2 and 3, with motif 2 exclusively present in the BcNF-YA subfamily, suggesting its potential role as a constituent of the CBF domain (Figure 1a). The BcNF-YB subfamily consistently harbored motif 1, except for BcNF-YB2, BcNF-YB14, and BcNF-YB15, which also contain motif 6. In contrast, the BcNF-YC subfamily exhibited a unique motif, motif 7 or 1 (Figure 1a and Appendix A).

Notably, significant differences in the number of introns were observed among the members of the BcNF-Y family in flowering Chinese cabbage. Moreover, while *BcNF-YC7* and *BcNF-YC8* lacked introns, *BcNF-YA1* had the highest number of introns (n, 19). Most members had a substantial number of introns (Figure 1c, Appendix A). These findings suggest that the NF-Y family may exhibit diverse regulatory patterns and functional roles in flowering Chinese cabbage.

### 2.3. Phylogenetic Analysis and Chromosomal Distribution

To classify the BcNF-Y family members into distinct subfamilies, a neighbor-joining phylogenetic tree was constructed using the rice and *Arabidopsis* NF-Y protein sequences (Appendix A). The phylogenetic analysis helped us predict the function of BcNF-Y by examining the relationship between *NY-F* genes from different species. The results showed three BcNF-Y subfamilies. NF-YA exhibited noticeable differences from the other two subfamilies, while NF-YB and NF-YC showed closer evolutionary relationships (Figure 2a).

The chromosomal localization of the 49 *BcNF-Y* genes in flowering Chinese cabbage was visualized using TBtools software (V1.0987). Based on their positions on chromosomes 1–10 (from top to bottom) and their classification into subfamilies, they were designated as follows: NF-YA subfamily (*BcNF-YA1* to *BcNF-YA17*), NF-YB subfamily (*BcNF-YB1* to *BcNF-YB20*), and NF-YC subfamily (*BcNF-YC1* to *BcNF-YC12*) (Figure 2b). The findings revealed an uneven distribution of *BcNF-Y* genes on the chromosomes, with chromosome A04 containing only one *BcNF-Y* gene and chromosome A09 harboring 10 *BcNF-Y* genes.

### 2.4. Gene Duplication and Collinearity Analysis

Analyzing the collinearity between genes provides valuable insights into gene family expansion and duplication events. In the case of *BcNF-Y* genes, collinearity analysis revealed 36 duplicated events within the gene family, with chromosome 3 showing the highest number of duplication events (Figure 3a). Among these duplicates, only *BcNF-YC5* and *BcNF-YC6* occurred on the same chromosome, while the rest occurred on different chromosomes (Figure 3a). The expansion of the *BcNF-Y* gene family is primarily attributed to segmental duplications.

Collinearity between genes often indicates the presence of homologous sequences that may share similar functions. By examining the collinearity between *BcNF-Y* and *AtNF-Y* genes and performing BLASTP and phylogenetic comparisons, the homologous genes of *BcNF-Y* in *Arabidopsis* were identified. The results showed that all 43 *BcNF-Y* genes have homologs in *Arabidopsis*. Specifically, *BcNF-YB4*, *BcNF-YB7*, and *BcNF-YB20* correspond to two homologous genes (*AtNF-YB8/10*) in *Arabidopsis*, while *BcNF-YA11* corresponds to two homologous genes (*AtNF-YA5/6*) (Figure 3b, Table 1). Understanding the homology between *BcNF-Y* and *AtNF-Y* contributes to better comprehending the function of *BcNF-Y* genes, considering the extensive studies conducted on *Arabidopsis AtNF-Y* genes.

### 2.5. Analysis of BcNF-Y Gene Expression in Different Periods and Tissues

To explore the expression patterns of *BcNF-Y* genes throughout the growth and development of flowering Chinese cabbage, transcriptome analysis was conducted using data from various developmental stages, including three leaf, bud emergence, and flowering stages [2]. The results showed diverse expression patterns of *BcNF-Y* genes during the developmental stages of flowering Chinese cabbage (Figure 4a). Among the 18 *BcNF-Y* genes analyzed (11 *BcNF-YA* and 7 *BcNF-YB*), their expression levels were upregulated during the three-leaf stage. Additionally, 15 *BcNF-Y* genes (3 *BcNF-YA*, 5 *BcNF-YB*, and 7 *BcNF-YC*) showed increased expression levels during the bud emergence stage, while 11 *BcNF-Y* genes (2 *BcNF-YA*, 4 *BcNF-YB*, and 5 *BcNF-YC*) exhibited elevated expression levels during the flowering stage (Figure 4a). Specifically, *BcNF-YA* genes exhibited high expression during the seedling stage, while *BcNF-YC* genes demonstrated higher expression levels, specifically during flower stalk development. These findings indicate that *BcNF-YA* may play a more prominent role during the seedling stage, while *BcNF-YC* appears to be closely associated with stalk and flower development.

Differential gene expression across different developmental stages and in different tissues often indicates their distinct functions. To gain further insight into the role of NF-Y in the growth and development of flowering Chinese cabbage and identify potential *BcNF-Y* genes associated with bolting and flowering, qRT-PCR analysis was performed on 27 *BcNF-Y* genes across eight stages (cotyledon, first leaf, second leaf, third leaf, fourth leaf, bolting, bud emergence, rapid bolting, and flowering) and three tissues (root, stem tip, and leaf). The findings revealed that during bud emergence and fast bolting, 18 genes were upregulated in specific tissues (Figure 4b). In the bud emergence, two *BcNF-Y* genes exhibited elevated expression in the stem apex, while seven genes showed increased expression in the leaves. Additionally, nine genes were upregulated in the root during bud emergence or rapid bolting stage. Notably, *BcNF-YA* showed higher expression in the leaves and roots than in the stem apex. Among the genes highly expressed in the stem apex, five of the six *BcNF-Y* genes belonged to the BcNF-YC subfamily, whereas the BcNF-YB subfamily members exhibited no specific expression trends (Figure 4b). These findings indicate that the BcNF-YC subfamily may play a crucial role in the bolting process of flowering Chinese cabbage, while BcNF-YA may primarily contribute to leaf and root development.

Overall, the findings indicate that a significant number of *BcNF-Y* genes were highly expressed during the critical stages of flowering and bolting, suggesting their key roles in the development of flowering Chinese cabbage.

### 2.6. BcNF-Y Gene Expression Analysis in Flowering Chinese Cabbage under Different Exogenous Hormone Spraying Conditions

To investigate the influence of the *NF-Y* gene family on GA-mediated bolting of flowering Chinese cabbage, we analyzed the transcript levels of *NF-Y* gene family members following exogenous spraying with GA3 and PAC (a GA inhibitor). After PAC treatment, substantial upregulation in transcript levels was observed for all members of the NF-YA and NF-YC subfamilies. Conversely, no discernible response to the GA treatment was observed among the members (Figure 5).

NF-YB subfamily members exhibited distinct expression patterns (Figure 5). *BcNF-YB5*, *BcNF-YB15*, and *BcNF-YB20* exhibited significantly elevated transcript levels after PAC treatment, whereas their expression levels were unaffected by GA treatment. However, *BcNF-YB14* showed the opposite trend, with significantly increased expression after GA treatment and no change in expression after PAC treatment (Figure 5). *BcNF-YB10* responded to GA3 and PAC treatments, with GA3 treatment significantly reducing its expression levels and PAC treatment significantly increasing its transcript levels (Figure 5).

### 2.7. Interactions of DELLAs with NF-Y Subunits and Inter-Subunit Interactions

DELLA proteins have crucial regulatory functions in the GA signaling pathway and are essential for plant development. To determine the interactions between the BcNF-Y and BcDELLA proteins in flowering Chinese cabbage, yeast two-hybrid experiments were conducted. The experiment revealed that BcNF-YA8, BcNF-YB14, BcNF-YB20, and BcNF-YC5 interacted with BcRGA1, while BcNF-YA12 and BcNF-YB15 did not interact with BcRGA1 (Figure 6a).

To determine the interactions among BcNF-Y subfamily members in flowering Chinese cabbage, yeast two-hybrid experiments were conducted. The results showed that BcNF-YA8 and BcNF-YA12 did not interact with BcNF-YB14, BcNF-YB15, or BcNF-YB20 (Figure 6b). However, BcNF-YC5, BcNF-YC5, and BcNF-YC5 interacted with BcNF-YA8, BcNF-YA12, and BcNF-YB15, respectively, suggesting their potential involvement in complex formation in flowering Chinese cabbage.

### 2.8. Subcellular Localization and Transcriptional Activation Analysis of NF-Y Genes

To further elucidate the functions of the BcNF-Y family, selected *NF-Y* genes were analyzed for their subcellular localization. The results show that BcNF-YA8-GFP and BcNF-YA12-GFP fluorescence was exclusively observed in the nucleus, indicating their localization within the nucleus (Figure 7a). In contrast, the BcNF-YB14-GFP, BcNF-YB20-GFP, BcNF-YC5-GFP, and BcNF-YB15-GFP fusion proteins exhibited fluorescence in the cell membrane and nucleus, indicating that these six NF-Y gene family members were localized in the cell membrane and nucleus of flowering Chinese cabbage (Figure 7a).

Selected members were subjected to transcriptional activity analysis to assess the transcriptional activation abilities of BcNF-Y gene family members. The coding sequence (CDS) of BcNF-Y was fused to the GAL4 DNA-binding domain (GAL4DB) to generate an effector construct, while an empty vector and a VP16 served as the negative and positive controls, respectively (Figure 7b). The results showed that GAL4DB-BcNF-YB15, GAL4DB-BcNF-YB20, and GAL4DB-BcNF-YA12 exhibited significantly enhanced relative luciferase activity than in the negative control, indicating that GAL4DB-BcNF-YB15, GAL4DB-BcNF-YB20, and GAL4DB-BcNF-YA12 may function as transcriptional activators in flowering Chinese cabbage (Figure 7c). Conversely, GAL4DB-BcNF-YA8 and GAL4DB-BcNF-YC5 exhibited significantly decreased relative luciferase activities, suggesting that GAL4DB-BcNF-YA8 and GAL4DB-BcNF-YC5 may function as transcriptional repressors (Figure 7c). GAL4DB-BcNF-YB14 exhibited no significant difference in relative luciferase activity, indicating that the full-length BcNF-YB14 protein cannot regulate transcription.

## 3. Discussion

The NF-Y transcription factor family plays a crucial role in various plant growth and development stages, and extensive research has been conducted on various plant species [32,33,34,35]. In this study, we identified 49 *BcNF-Y* genes, consisting of 17 *BcNF-A*, 20 *BcNF-YB*, and 12 *BcNF-YC* genes. Furthermore, investigation of the expression patterns of these *BcNF-Y* genes across distinct developmental stages and tissues of flowering Chinese cabbage provided insight into their potential roles in plant development. Additionally, specific NF-Y proteins that interacted with BcRGA1 were identified, and subsequent subcellular localization and transcriptional activation assays were used to elucidate their functional properties. These comprehensive findings of this study lay the groundwork for further functional validation of *BcNF-Y* genes and the identification of key regulators involved in developing flower stalks in flowering Chinese cabbage.

Compared to *Arabidopsis*, a closely related cruciferous family member, flowering Chinese cabbage exhibited an expansion of the *NF-Y* gene family with the addition of 13 members [36]. The identification of fragment duplication events supports this expansion. Each subfamily exhibited distinct structural features, conserved domains, and motif compositions, indicating the evolutionary conservation of *NF-Y* genes in flowering Chinese cabbage. Introns play a critical role in splicing processes [37]. Alternative splicing enables a single gene to generate different mature or messenger RNAs, thereby expanding the proteome of an organism [38]. Furthermore, the number of introns in *BcNF-Y* family members was higher than in other plant species, such as alfalfa, castor, and peach. This suggests a potentially significant regulatory pattern and functional role for *BcNF-Y* genes in flowering Chinese cabbage [34,35,39].

NF-Y factors typically form heterotrimeric complexes that recognize and bind to CCAAT sequences to regulate transcription [40,41]. In the case of NF-YB and NF-YC, they form dimers in the cytoplasm and then translocate to the nucleus, where they recruit NF-YA to assemble a functional heterotrimeric complex [9]. There was no interaction between BcNF-YA and BcNF-YB in the yeast two-hybrid system, which is consistent with observations in *Arabidopsis* [42]. The complexity of NF-Y subunit interactions allows for better targeting of CCAAT boxes and fine-tuning of gene regulation [36].

In flowering Chinese cabbage, 18 *BcNF-Y* genes were found to be significantly upregulated during the bud emergence and fast-bolting stages, indicating their potential involvement in flowering stalk development. In *Arabidopsis*, AtNF-YC3, AtNF-YC4, and AtNF-YC9 are important factors in the CO-mediated flowering pathway and can redundantly regulate GA- and ABA-mediated seed germination [29]. Similarly, AtNF-YA2, AtNF-YB2, and AtNF-YC9 can bind to NFYBE, leading to the activation of SOC1 transcription and regulation of flowering in *Arabidopsis* [28]. Moreover, silencing the rose homologs of *AtNF-YC9* and *RhNF-YC9* resulted in reduced GA accumulation and significant downregulation of genes involved in cell expansion, inhibiting petal expansion [30]. In flowering Chinese cabbage, *BcNF-YC10* is a homolog of *AtNF-YC9* and exhibits significant upregulation in the stem tip during the bud emergence stage. Based on this observation, we hypothesized that *BcNF-YC10* is an important candidate gene involved in developing flowering Chinese cabbage flower stalks. In *Arabidopsis*, overexpression of *AtNF-YA2* leads to a delay in flowering, and miR169 regulates the stress-mediated flowering process and lateral root development by regulating AtNF-YA2 [43]. In flowering Chinese cabbage, *BcNF-YA6* is a homolog of *AtNF-YA2*, exhibiting significantly high expression in roots at the bud emergence stage. This suggests that *BcNF-YA6* may play a functional role comparable to *AtNF-YA2*. Therefore, we hypothesized that *BcNF-YA6* is crucial in flowering and root development in flowering Chinese cabbage. Further studies on the overexpression and knockdown of *BcNF-Ys* could help elucidate their roles in flowering Chinese cabbage bolting and flowering.

GA is a well-known regulator of flowering and bolting in flowering Chinese cabbage. Treating with exogenous GA3 can help accelerate bolting and flowering processes while promoting stem elongation [44]. Although most *BcNF-Y* family members showed increased expression following treatment with PAC (a GA synthesis inhibitor), their responses to GA treatment were insignificant. However, *BcNF-YB14* responded to GA and PAC, with its expression significantly upregulated in the leaves during the bolting stage. This suggests that *BcNF-YB14* may be involved in GA-mediated bolting in flowering Chinese cabbage plants. In flowering Chinese cabbage, DELLA proteins function as negative regulators of the GA signaling pathway, influencing the bolting and flowering processes. One such DELLA protein in flowering Chinese cabbage is BcRGA1 [4]. Using yeast two-hybrid analysis, the interactions between BcRGA1 (a DELLA protein) and specific BcNF-Y proteins, including BcNF-YA8, BcNF-YB14, BcNF-YB20, and BcNF-YC5 were identified. During the bolting stage, *BcNF-YA8* was significantly expressed in the leaves, while *BcNF-YB14* was highly expressed in stem tips during the flowering stage, and *BcNF-YB20* was highly expressed in roots during the fast bolting stage. Transcriptional activity analysis revealed that BcNF-YA8, BcNF-YB20, and BcNF-YC5 have regulatory roles in downstream transcription. However, BcNF-YB14 did not exhibit transcriptional regulation, possibly because of the presence of a transcriptional repressor domain in the full-length protein. These findings suggest that these four identified *BcNF-Y* genes likely participate in GA-mediated flowering and bolting processes in flowering Chinese cabbage.

Transcriptional regulation mediated by NF-Y involves a complex mechanism. In *Arabidopsis*, CO can substitute NF-YA to form a CO/NF-YB/NF-YC trimer, subsequently binding to and regulating the FT promoter [45]. The entire heterotrimeric complex can recruit additional transcription factors, thereby modulating its binding affinity for CCAAT boxes [46,47]. Moreover, a pair of AtNF-Y protein complexes (NFY and NF-CO) are in close proximity to each other and simultaneously bind to two CCAAT boxes located >5.3 kb apart on the FT promoter, regulating flowering [25,45]. The transcriptional regulatory effects of NF-Y on downstream genes primarily rely on intact NF-Y complexes. However, the identification of intact and active NF-Y complexes remains challenging. Nevertheless, we did not identify any candidate combinations of the NF-Y complexes. Advancements in bioinformatics and more accurate protein interaction predictions can expedite the exploration of NF-Y complex compositions and facilitate functional studies of NF-Y complexes.

## 4. Materials and Methods

### 4.1. Identification of the BcNF-Y Gene Family

AtNF-Ys proteins sequence of *Arabidopsis* were derived from a previous report [36]. The protein sequences of AtNF-Ys were used as queries and compared to the *Brassica campestris* genome using BLASTP with an e-value cutoff of 1 × 10^−5^ and an identity threshold of >40%. On the other hand, the hidden Markov model (HMM) profile for PF02045 and PF00808 were downloaded from the Pfam database (https://www.ebi.ac.uk/interpro/, accessed on 7 October 2022), which was used to conduct another search with HMMER v3.3.1 (e-value 1 × 10^−5^). The non-redundant protein sequences obtained from both methods were further analyzed using the NCBI Conserved Domain (CD) tool (https://www.ncbi.nlm.nih.gov/Structure/bwrpsb/bwrpsb.cgi, accessed on 7 October 2022) and the SMART database (https://smart.embl.de/, accessed on 7 October 2022) for confirmation.

### 4.2. Analysis of Gene Structure, Domains, and Conserved Motifs

Conserved motifs were identified using the online program MEME v5.4.1 with the find motifs parameter set to 10, while other parameters were kept as default (https://meme-suite.org/meme/; accessed on 7 October 2022). The SMART database (https://smart.embl.de/, accessed on 7 October 2022) was used to identify conserved domains with the default parameters. TBtools software (V1.0987) was used to conduct the structural visualization analysis [48].

### 4.3. Phylogenetic Analysis

*NF-Y* gene sequences from *Arabidopsis* (*AtNF-Ys*) and rice (*OsNF-Ys*) were obtained from previous studies [36,49]. To align all NF-Y, we used the L-INS-I method of the MAFFT sequence alignment program [50]. A neighbor-joining phylogenetic tree was then constructed using MEGA11, and the reliability of the tree was assessed using a bootstrap test with 1000 replicates [51].

### 4.4. Chromosomal Locations, Synteny Analysis

The chromosomal location of *BcNF-Y* genes was determined using TBtools software (V1.0987). To analyze the synteny of *NF-Y* genes, we employed MCScanX with the default parameters in TBtools (V1.0987) to perform synteny analysis of *NF-Y* genes was performed in *Arabidopsis*, rice, *Brassica campestris*, and within *Brassica campestris* [52]. Homologous genes between *Brassica campestris* and *Arabidopsis thaliana* were identified, and synteny analysis was conducted. The results of the analysis were visualized using TBtools.

### 4.5. Plant Materials and Treatments

The “youIv501” variety of flowering Chinese cabbage plants was cultivated in a daylight greenhouse located at the Department of Facility Horticulture, South China Agricultural University, using a substrate potting technique. To initiate the experiment, the seeds were sterilized and placed on Petri dishes containing a moist filter for 1 d. Following germination, the seeds were transferred to cavity trays and then transplanted into seedling pots filled with a substrate mixture composed of peat, vermiculite, and perlite (in a ratio of 3:1:1) when the seedlings reached the three-leaf stage. Tissue samples were collected at various seedling developmental stages of the seedlings, including the cotyledon, two-leaf, three-leaf, four-leaf, bolting, bud emergence, fast bolting, and flowering stages. Stem tips, roots, and leaf tissues were collected at each stage. For the exogenous hormone treatment, GA3 (200 mg/L) and PAC (GA synthesis inhibitor) (10 mg/L) were sprayed onto the seedlings at the three-leaf stage. Stem tip samples were collected 12 h after the hormone treatment. Each treatment was replicated three times with 20 seedlings per treatment. All collected samples were stored at −80 °C and later used for RNA extraction.

### 4.6. RNA Extraction and qRT-PCR

Total RNA was extracted from the sample using the Eastep^®^ Super Total RNA Extraction Kit, and genomic DNA was removed. The cDNAs were prepared using Hiscript QRT SuperMix (Vazyme, Nanjing, China). The real-time PCR (qRT-PCR) were performed using the ChamQ SYBR Color qPCR Master Mix (Vazyme, Nanjing, China). Primers were designed using the NCBI BLAST tool with glyceraldehyde-3-phosphate dehydrogenase (GAPDH) as the internal reference gene. The relative expression levels were determined using the 2^−∆∆Ct^ method. qRT-PCR was conducted following the previously described protocol [44]. Appendix A lists the primers used for the qRT-PCR analyses.

### 4.7. Yeast Two-Hybrid Assay

Full-length coding sequences (CDS) of *BcNF-Ys* and *BcRGA1* genes were cloned into pGADT7 and pGBKT7 vectors, respectively. The individual vectors were co-transformed into yeast cells using the Y2HGold Competent Cells (WEIDI, Shanghai, China). The co-transformed yeast cells were then plated on a selective medium lacking leucine and tryptophan (SD/-Leu/-Trp) to ensure the presence of both plasmids. The yeast transformants were then plated on selective media lacking leucine, tryptophan, histidine, and adenine (SD/-Trp/-Leu/-His/-Ade) to assess protein interactions. X-α-Gal medium was used to visualize and confirm the interactions, enabling the identification of blue and white colonies. Appendix A shows all yeast two-hybrid primers.

### 4.8. DLR Assay

The transient transcriptional activity assay was conducted on *Nicotiana benthamiana* leaves using a previously described method [53,54]. The full-length coding sequences (CDS) of *BcNF-Ys* genes were cloned into the GAL4DB vector to generate the GAL4DB-BcNF-Ys vector. For the experimental setup, GAL4DB-VP16 was used as the positive control, GAL4DB-Empty as the negative control, and GAL4DB-BcNF-Ys as the test vector. *Agrobacterium tumefaciens* strains harboring different plasmids were infiltrated into tobacco leaves, followed by a 48 h incubation period. Determination the luciferase/Renilla luciferase (Luc/Ren) ratio, enabling the analysis of transcription-al activity [6]. Appendix A lists all DLR assay primers.

### 4.9. Subcellular Localization

The full-length CDS sequence of *BcNF-Ys* without the stop codon was cloned into the pSUPER1300 vector. The pSUPER1300 empty vector was used as a control. The recombinant vectors and a localization signal (DsRed) were introduced into Agrobacterium tumefaciens GV3101 and co-injected into tobacco leaves. After two days, the fluorescence signal was observed using laser scanning confocal microscopy (Axioimager.D2). Appendix A lists all subcellular localization primers.

## Figures and Tables

**Figure 1 ijms-24-11898-f001:**
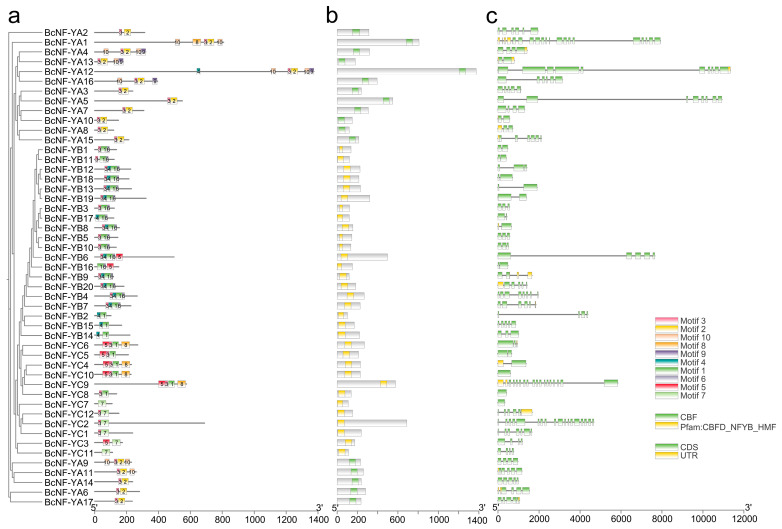
Schematic representation of protein and gene structure. (**a**) Distribution of conserved motifs in BcNF-Y proteins; different colors represent the 10 conserved domains identified. (**b**) Each domain is visually depicted as a colored box. (**c**) Gene structure of *BcNF-Y* genes.

**Figure 2 ijms-24-11898-f002:**
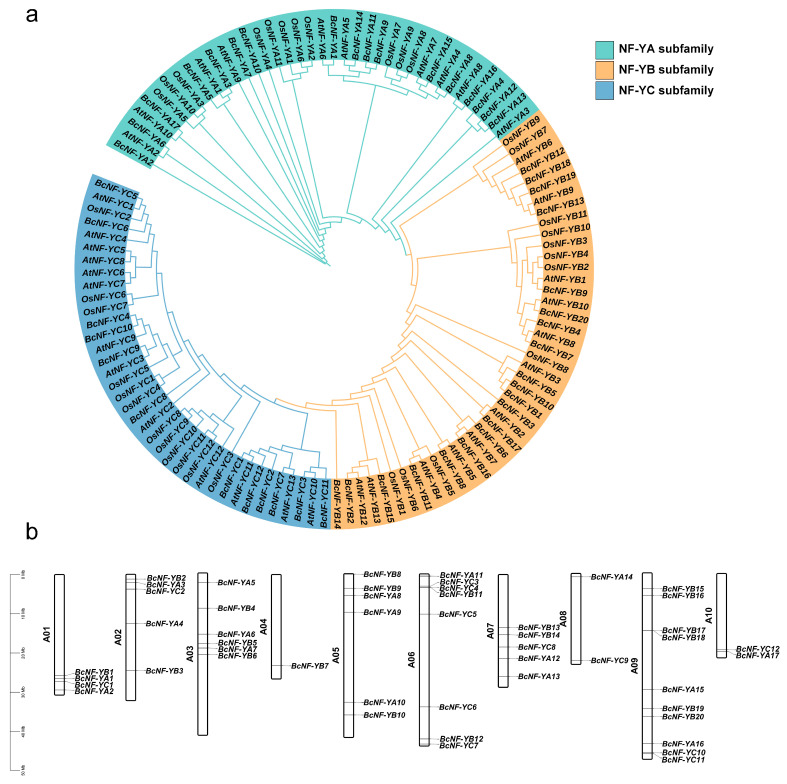
Phylogenetic analysis of NF-Y proteins in *Brassica campestris*, rice, *Arabidopsis*, and chromosomal localization of *BcNF-Y*. (**a**) Unrooted neighbor-joining phylogenetic tree of NF-Y in flowering Chinese cabbage, *Arabidopsis*, and rice. The phylogenetic tree is divided into three subfamilies, and different colors are used to distinguish members of each subfamily. (**b**) Chromosomal distribution of 49 *BcNF-Y* genes in flowering Chinese cabbage.

**Figure 3 ijms-24-11898-f003:**
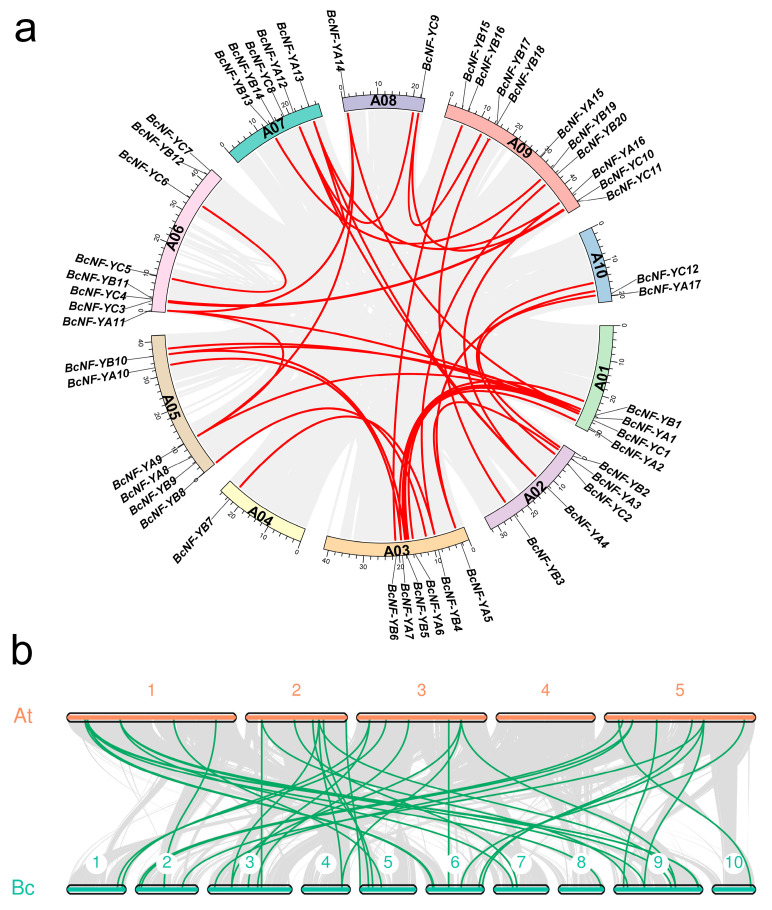
Gene duplication and collinearity analysis in *Brassica campestris*. (**a**) Duplication analysis of *BcNF-Y* genes on different chromosomes. The red lines connect *BcNF-Y* genes with a collinearity relationship. (**b**) Collinearity analysis of *NF-Ys* from *Brassica campestris* and *Arabidopsis*. The green lines indicate the presence of a collinearity relationship between NF-Y genes.

**Figure 4 ijms-24-11898-f004:**
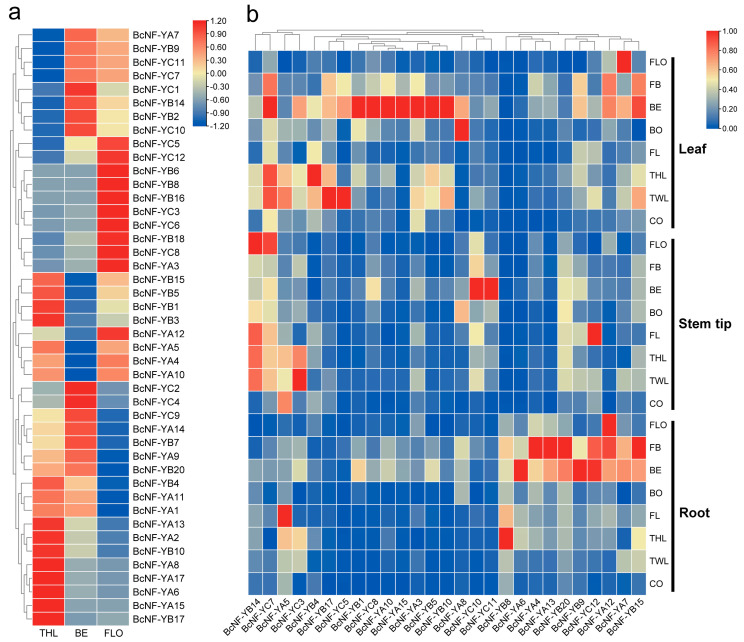
Tissue-specific expressions of *BcNF-Y* genes in flowering Chinese cabbage. (**a**) Heatmap construction based on the transcripts per million (TPM) values. (**b**) Heatmap of *BcNF-Y* gene expression patterns in different periods and tissues. CO, cotyledon stage; TWL, two-leaf stage; THL, three-leaf stage; FL, four-leaf stage; BO, bolting stage; BE, bud emergence stage; FB, fast bolting stage; FLO, flowering stage.

**Figure 5 ijms-24-11898-f005:**
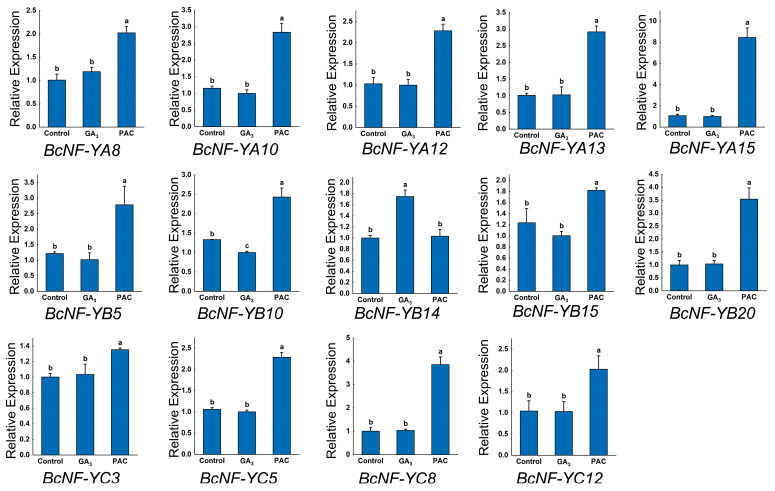
Effects of GA3 and PAC treatments on *BcNF-Ys* expression in flowering Chinese cabbage. Data represent the mean ± standard error for three biological experiments. Different letters (a, b, and c) indicate significant differences (*p* < 0.05).

**Figure 6 ijms-24-11898-f006:**
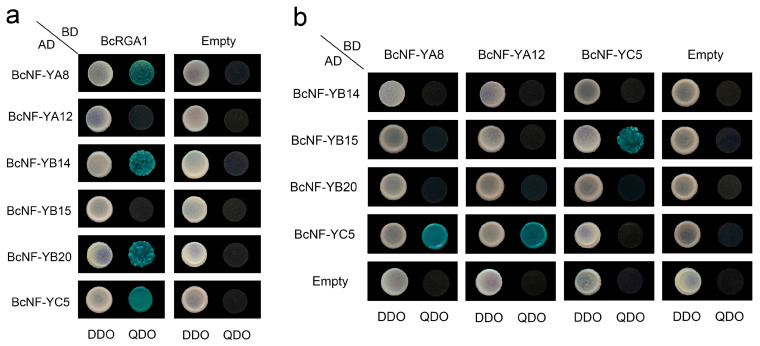
Protein interaction determination using yeast two-hybrid. (**a**) Yeast two-hybrid assay was employed to investigate protein interactions between BcNF-Ys and BcRGA1. (**b**) Interactions among NF-Y subfamily members at the protein. DDO and QDO represent SD/-Trp-Leu medium and SD/-Trp-His-Leu-Ade medium, respectively. Positive bacteria were stained using X-α-Gal. The combinations containing AD-empty or BD-empty were used as negative controls.

**Figure 7 ijms-24-11898-f007:**
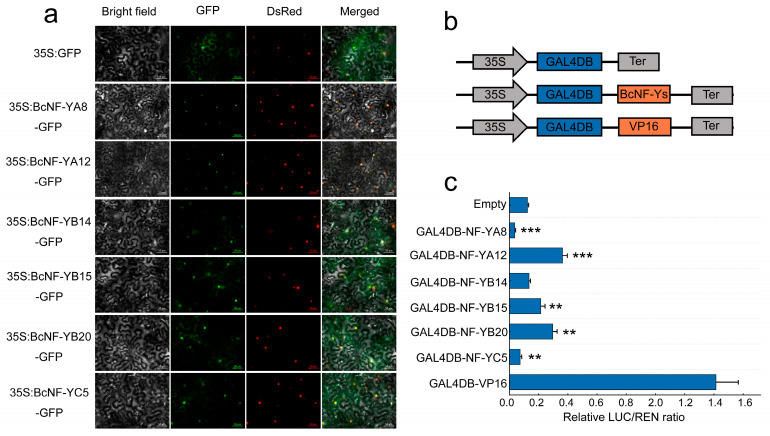
Subcellular localization and transcriptional activation analysis. (**a**) Localization of BcNF-Ys within the cells of *Nicotiana benthamiana* was examined. (**b**) Schematic illustration for transcriptional activity assay. (**c**) The relative luciferase (LUC) activities observed in *Nicotiana benthamiana* leaves indicate the transcriptional activation potential of BcNF-Y proteins. ‘**’ indicates *p*-value < 0.01, and ‘***’ indicates *p*-value < 0.001, by one-way ANOVA analysis with Tukey’s HSD test.

**Table 1 ijms-24-11898-t001:** Information of the *BcNF-Y* genes family in flowering Chinese cabbage.

Gene Name	CDS (bp)	Length (AA)	pI	MW (Da)	Homologs in *Arabidopsis*
*BcNF-YA1*	2430	809	9.36	34,401.98	*AtNF-YA6*
*BcNF-YA2*	945	314	9.87	13,514.28	*AtNF-YA2*
*BcNF-YA3*	720	239	9.54	28,766	*AtNF-YA1*
*BcNF-YA4*	963	320	9.99	20,085.81	*AtNF-YA3*
*BcNF-YA5*	1653	550	7.16	155,711.3	*AtNF-YA1*
*BcNF-YA6*	846	281	9.74	26,426.97	*AtNF-YA2*
*BcNF-YA7*	927	308	6.87	44,034.04	*AtNF-YA9*
*BcNF-YA8*	360	119	9.51	24,406.47	*AtNF-YA4*
*BcNF-YA9*	696	231	9.61	27,110.81	*AtNF-YA5*
*BcNF-YA10*	450	149	6.99	88,896.45	*AtNF-YA9*
*BcNF-YA11*	783	260	9.04	35,419.5	*AtNF-YA5/6*
*BcNF-YA12*	4125	1374	6.32	26,437.07	*AtNF-YA3*
*BcNF-YA13*	540	179	8.99	34,091.57	*AtNF-YA3*
*BcNF-YA14*	717	238	9.68	30,882.8	*AtNF-YA5*
*BcNF-YA15*	645	214	6.36	61,377.49	*AtNF-YA7*
*BcNF-YA16*	1188	395	9.95	17,017.99	*AtNF-YA8*
*BcNF-YA17*	711	236	9.9	25,671.99	*AtNF-YA10*
*BcNF-YB1*	414	137	7.07	14,745.38	*AtNF-YB3*
*BcNF-YB2*	309	102	6.91	25,881.06	*AtNF-YB12*
*BcNF-YB3*	372	123	5.06	23,121.14	*AtNF-YB2*
*BcNF-YB4*	804	267	7.76	13,714.15	*AtNF-YB8/10*
*BcNF-YB5*	438	145	5.1	25,587.71	*AtNF-YB3*
*BcNF-YB6*	1497	498	8.84	19,813.36	*AtNF-YB7*
*BcNF-YB7*	684	227	5.04	15,948.19	*AtNF-YB8/10*
*BcNF-YB8*	468	155	4.27	24,485.01	*AtNF-YB5*
*BcNF-YB9*	357	118	5.96	25,267.27	*AtNF-YB1*
*BcNF-YB10*	408	135	8.92	13,174.68	*AtNF-YB3*
*BcNF-YB11*	369	122	4.8	11,670.12	*AtNF-YB4*
*BcNF-YB12*	681	226	5.01	56,994.71	*AtNF-YB2*
*BcNF-YB13*	693	230	7.07	15,677.32	*AtNF-YB9*
*BcNF-YB14*	666	221	8.8	29,546.27	*AtNF-YB11*
*BcNF-YB15*	510	169	9.61	25,962.1	*AtNF-YB13*
*BcNF-YB16*	456	151	6.91	14,458.08	*AtNF-YB7*
*BcNF-YB17*	363	120	5.21	13,011.72	*AtNF-YB2*
*BcNF-YB18*	648	215	6.65	17,694.84	*AtNF-YB9*
*BcNF-YB19*	969	322	10.11	12,238.18	*AtNF-YB9*
*BcNF-YB20*	552	183	5.34	29,312.26	*AtNF-YB8/10*
*BcNF-YC1*	717	238	7.12	63,811.16	*AtNF-YC11*
*BcNF-YC2*	2070	689	5.7	35,706.48	*AtNF-YC11*
*BcNF-YC3*	525	174	6.22	23,990.09	*AtNF-YC10*
*BcNF-YC4*	693	230	5.72	12,891.28	*AtNF-YC9*
*BcNF-YC5*	636	211	6.7	17,457.36	*AtNF-YC1*
*BcNF-YC6*	813	270	4.74	19,047.31	*AtNF-YC4*
*BcNF-YC7*	330	109	8.49	17,259.75	*AtNF-YC13*
*BcNF-YC8*	417	138	8.9	26,409.45	*AtNF-YC2*
*BcNF-YC9*	1734	577	6.93	75,108.26	*AtNF-YC9*
*BcNF-YC10*	687	228	6.13	12,824.61	*AtNF-YC9*
*BcNF-YC11*	339	112	5.1	25,421.53	*AtNF-YC10*
*BcNF-YC12*	459	152	9.1	19,965.36	*AtNF-YC11*

## Data Availability

All important data are included in the article and Appendix A.

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
