# Peer review of "Genome-Wide Identification of the NF-Y Gene Family and Their Involvement in Bolting and Flowering in Flowering Chinese Cabbage"

_ijms, 2023, doi:10.3390/ijms241511898_

Round 1
Reviewer 1 Report
In this manuscript authors have identified and described the NF-Y gene family in Chinese Cabbage, which allow obtainig new information in different species that it is interesting for evolutive comparison and stablishing differences/similarities in regulation.
The work is well written and experiments are well designed, and conclusions are almost totally in coherence with data, so I want first to congratulate authors.
There are some points I want to highlight to authors:
1) The quality of figures 1 and 2 must be improved. Althought the number of gene family genes is high, there are a lot of manuscripts with more genes (even hundred of genes) and the phylogenetic trees are legible. Tree lines must be thicker and letter size also bigger. Additionally, inside the colored area, text indicating the subfamily (NF-YA, NF-YB and NF-YC) can be added for facilitating the interpretation.
2) In all the manuscript several comparition with other species, specially Arabidopsis, are indicated. Althougt is expected (and necessary) this kind of comparisons, data from the others species must be indicated explicitly at least as supplementary information. For instance, in point 2.2 and in discussion the number of introns is commented and compared with other species, however there are no data about the number of introns in homologous genes in Arabidopsis or other species. I am sure auhors have that information, as they commented it in the manuscript, so it would be easy to add it (with the corresponding reference).
3) In table 1 I found that homology relation is not well interpreted. It can not be possible that a gene of the family have no homolog in Arabidopsis, taken into account that homology stablish an evolutive relation and all the genes in the family have that evolutive relation. I understand that authors wanted to present the unambiguous ortolog relation between both species genes. Anyway, the gene family seems to change fast some parts of the protein and several duplication events have ocurred, so maybe a more detailed comparison of exon sizes and triplet-codon alignment would be necessary to more clearly identify ortologs and paralogs. Alternatively, authors could simply put in the table the “closest” homolog (instead of just homolog) and us phylogeny to assign it (then all genes must have an homolog).
- Detail in lines 170-171: listing genes will be more logical if elements followed “number” order in addition to alphabetical order (YB10 is listed before YB5 and YC12 before YC2).
4) Also related with comparisons, in lines 196-210 expression of different NF-Y genes is commented. I think it would be interesting to add comparison with behaviour of its ortolog genes in Arabidopsis. If the Arabidopsis ortologs have the same expression pattern is interesting because it reflects a possible conserved mechanism, but it is also interesting if expression pattern is not the same or the NF-Y genes with the same pattern are not the direct ortologs as it can be for adaptation to different flowering conditions or due to function flexibility of homologs, for instance. I think it is something easy to do and adds additional interesting information.
5) The number of NF-Y genes described in some experiments are not well explained of seems a little confusing. In line 185 it say “among the 15 BcNF-Y genes” but in Figure 4 there are more than 15. As authors use published data it expected they search for all NF-Y genes, though after explore data see that some of the genes do not present changes in expression. This kind of incoherence between number in the text and number of genes in figures is detected in a few cases.
6) In the case of protein interaction, is also not well explained how many genes have been tested (as primer pairs in suplementary data are more than genes in Figure 6). If many more combination have been tested, although with negative results, data must be showed (as supplementary information), and in the text negative results must be better described or that all combination have been tested but only positive results are shown. From primers info I guess (not cheked) authors have selected those genes with changes in the explored conditions, however this can be a mistake, as no changes in expression does not means is not necessary to form the protein complex. I think that interactions between subunits must be tested all-vs-all.
Additionally, is not well supported why DELLA interaction is tested. In the introduction and before point 2.7, all information connected GAs and NF-Y genes through transcriptional regulation. I think it must indicated reference of DELLA and NF-Y interaction in Arabidopsis as argument to test it in Cabbage .
The same occurs with interaction tested in yeast. It is not clear how many genes have been tested. It is indicated that “selected NF-Y genes were analyzed” but it is not clear: in fig 6.a there are 6 genes against RGA1 but in fig6.b 4 genes (B14,B15,B20,C5) against 3 genes. Negative results in this case is useful information, not testing the interaction is not the same that doing it and find they do not interact.
7) In discussion, I suggest to be more realistic in general. For instance, in lines 318-320, authors can not say that genes have a crucial role in flowering stalk develpoment just for correlation in pattern expression. Thousands of genes must have the same behaviour and not all of them have a “crucial role”. Authors can identify interesting correlations but by now it is only correlation.
8) In M&M I found some lack of information. For instance, it is necessary indicate the parameters used for each bioinformatic use: have used authors parameters “by defect” or they have changed some of them? For example, MEME software is sensitive to changes in motif size or minimal values and can give slightly different results. In the case of phylogeny, distances can be different depending of the sustitution model used. This kind of data are not well described.
9) I found weird the sentece in lines 144-145, as has no sense to say that based in their positions on chromosomes genes were assigned to three subfamily… Assignation to subfamilies must be based in phylogenetic relation and position on chromosome is not relevant in this case.
Minor point:
In my opinion, in lines 104-105, it would be more clear to use “to” instead or large dash: BcNF-YA1 to BcNF-YA17…
Nothing remarkable
Reviewer 2 Report
The manuscript “Genome-Wide Identification of the NF-Y Gene Family and their Involvement in Bolting and Flowering in Flowering Chinese Cabbage” described the comprehensive functional analysis of NF-Y genes in Chinese cabbage using critical conserved motif, conserved domain, gene structure, chromosomal location, synteny and phylogenetics analysis. Further, qRT-PCR, subcellular localization, yeast two-hybrid interaction and transcriptional activation activity were extensively studied and provide valuable information on the role NF-Y genes in flowering Chinese cabbage.
Minor errors
Page 1, Line 12, please italicize the scientific name Brassica campestris L. ssp. Chinensis var. utilis Tsen et Lee)
Page 2, Line 65, Arabidopsis, please italicize the scientific name
Page 7, Line 158, In the case of IBcNF-Y genes, why there is I before the gene name, are there any specific reasons?
Page 7, Line 168, BcNF-YB20, please italicize the gene name
Page 14, Line 398, ​​Brassica campestris, please italicize the scientific name
Page 15, Line 437, Nicotiana benthamiana, please italicize the scientific name
Page 15, Line 442, Line 449, Agrobacterium tumefaciens, please italicize the scientific name
Page 15, Line 433, X-alpha-Gal
Major concerns
1.What are the rationales that rice and Arabidopsis NF-Y family members were included in the phylogenetic tree analysis?
2. Line 412-416, For the exogenous treatment of the GA3 and PAC, different groups of BcNF-Y response and express differently in different stages and also behave differently in leave, stem tip, and root, what are the reasons authors specifically choose the three-leave stages and how did the authors quantify the expression level of the different plant tissue related genes after response to the hormone treatment?
